# Efficient Entropy for Policy Gradient with Multi-dimensional Action Space

**Yiming Zhang[1], Quan Ho Vuong[2], Kenny Song[3], Xiao-Yue Gong[4], Keith W. Ross[1,3]**
[1]New York University
[2]New York University Abu Dhabi
[3]New York University Shanghai
[4]Massachusetts Institute of Technology

## 1 Overview

This paper considers entropy bonus, which is used to encourage exploration in policy gradient. In the case of high-dimensional action spaces, calculating the entropy and its gradient requires enumerating all the actions in the action space and running forward and backpropagation for each action, which may be computationally infeasible. We develop several novel unbiased estimators for the entropy bonus and its gradient. We apply these estimators to several models for the parameterized policies, including Independent Sampling, CommNet, Autoregressive with Modified MDP, and Autoregressive with LSTM. Finally, we test our algorithms on a multi-hunter multi-rabbit grid environment. The results show that our entropy estimators substantially improve performance with marginal additional computational cost.

## 2 Entropy Bonus Approximation for Large Action Space

Consider an MDP with a $d$-dimensional action space $\mathcal{A} = \mathcal{A}_1 \times \mathcal{A}_2 \times \ldots \times \mathcal{A}_d$, and denote $\mathbf{a} = (a_1, \ldots, a_d)$ for an action in $\mathcal{A}$. To abbreviate notations, we write $p_\theta(\mathbf{a})$ for the parameterized policy $\pi_\theta(\mathbf{a}|s_t)$ and $\mathbf{a}_i$ for $(a_1, a_2, \ldots, a_i)$. We consider auto-regressive models whereby the sample components $a_i$, $i = 1, \ldots, d$ are sequentially generated. In particular, after obtaining $a_1, a_2, \ldots, a_{i-1}$, we will generate $a_i \in \mathcal{A}_i$ from some parameterized distribution $p_\theta(\cdot|\mathbf{a}_{i-1})$ defined over the one-dimensional set $\mathcal{A}_i$. After generating the distribution $p_\theta(\cdot|\mathbf{a}_{i-1})$, $i = 1, \ldots, d$ and the action components $a_1, \ldots, a_d$ sequentially, we then define $p_\theta(\mathbf{a}) = \prod_{i=1}^{d} p_\theta(a_i|\mathbf{a}_{i-1})$.

In policy gradient, we consider a set of parameterized policies $\pi_\theta(\cdot|s)$, $\theta \in \Theta$, and attempt to find a good $\theta$ within a parameter set $\Theta$. The parameters $\theta$ are updated by performing stochastic gradient ascent on the expected reward. One example of such an algorithm is REINFORCE (Williams & Peng (1991)) where an entropy bonus term is typically added to the gradient update to encourage exploration. However, in high-dimensional action space settings, calculating the entropy requires enumerating over the whole action space which is typically computationally infeasible.

### 2.1 Entropy Estimation

During training within an episode, for each state $s_t$, the policy generates an action $\mathbf{a}$. We refer to this generated action as the episodic sample. A crude approximation of the entropy bonus is:

$$H_\theta^{crude}(\mathbf{a}) = -\log p_\theta(\mathbf{a}) = -\sum_{i=1}^{d} \log p_\theta(a_i|\mathbf{a}_{i-1}).$$

This approximation is an unbiased estimate of $H_\theta$ but its variance is likely to be large.

We propose an alternative unbiased estimator for $H_\theta$ which only requires one action sample and accounts for the entropy along each dimension of the action space:

$$\widetilde{H}_\theta(\mathbf{a}) := -\sum_{i=1}^{d} \sum_{a \in \mathcal{A}_i} p_\theta(a|\mathbf{a}_{i-1}) \log p_\theta(a|\mathbf{a}_{i-1}) = \sum_{i=1}^{d} H_\theta^{(i)}(\mathbf{a}_{i-1})$$

where $H_\theta^{(i)}(\mathbf{a}_{i-1}) := -\sum_{a \in \mathcal{A}_i} p_\theta(a|\mathbf{a}_{i-1}) \log p_\theta(a|\mathbf{a}_{i-1})$, which is the entropy of $A_i$ conditioned on $\mathbf{a}_{i-1}$. This estimate of the entropy bonus is computationally efficient since for each dimension $i$, we need to obtain $p_\theta(\cdot|\mathbf{a}_{i-1})$, its log and gradient anyway during training.

We refer to this estimator as the *smoothed entropy estimator*. We were able to show theoretically that it can serve as a good proxy for the actual entropy in that it is an unbiased estimator, has the same minimum and maximum values when varying the model parameters, and in the special case when the model is a multivariate normal distribution, it is actually equal to the exact entropy.

## 2.2 Entropy Gradient Estimation

So far we have been looking at estimates of entropy. But the policy gradient update uses the gradient of the entropy rather than the entropy. As it turns out, the gradients of the estimators $H_\theta^{crude}(\mathbf{a})$ and $\widetilde{H}_\theta(\mathbf{a})$ are not unbiased estimates of the gradient of the entropy.

Analogous to the smoothed estimator for entropy, we can also derive a smoothed estimator for the gradient of the entropy.

**Theorem 1.** *If $\mathbf{a}$ is a sample from $p_\theta(\cdot)$, then*

$$\nabla_\theta \widetilde{H}_\theta(\mathbf{a}) + \sum_{i=1}^{d} H_\theta^{(i)}(\mathbf{a}_{i-1}) \nabla_\theta \sum_{j=1}^{i-1} \log p_\theta(a_j|\mathbf{a}_{j-1})$$

*is an unbiased estimator for the gradient of the entropy*

Note that this estimate for the gradient of the entropy is equal to the gradient of the smoothed estimate $\widetilde{H}_\theta(\mathbf{a})$ plus a correction term. We refer to this estimate of the entropy gradient as the unbiased gradient estimate.

## 3 Policy parameterization for efficient sampling

We will apply the estimators in the previous section to several models for the paramaterized marginal policies $p_\theta(a|\mathbf{a}_{i-1})$, $a \in \mathcal{A}_i$. In this discussion, we assume that the size of the one-dimensional action sets are equal, that is, $|\mathcal{A}_1| = |\mathcal{A}_2| = ... = |\mathcal{A}_d| = K$. To handle action sets of different sizes, we include inconsequential actions. We will consider the Independent Sampling (IS) baseline policy (Sukhbaatar et al. (2016)), MMDP Metz et al. (2017), and LSTM policy. We note the LSTM approach is an adaptation of sequence modeling in supervised machine learning (van den Oord et al. (2016)) to reinforcement learning and has also been proposed by Metz et al. (2017) and Bahdanau et al. (2016). See Figure 1 for details.

## 4 Experiment Results

We designed the hunters and rabbits to compare the different entropy estimators for three models discussed in the previous section as well as for the CommNet model (Sukhbaatar et al. (2016)). For each entropy approximation, the entropy weight $\beta$ was tuned to give the highest reward. The environment consists of an $n \times n$ grid. At the beginning of each episode, $d$ hunters and $d$ rabbits are randomly placed in the grid. The rabbits remain fixed in the episode, and each hunter can move to a neighboring square (including diagonal neighbors) or stay at the current square (i.e. $|\mathcal{A}| = 9^d$ actions). When a hunter enters a square with a rabbit, the hunter captures the rabbit and remains there until the end of the episode. The goal is for the hunters to capture the rabbits as quickly as possible.

Table 1 shows the performance of the IS, LSTM, MMDP and CommNet models with the different entropy estimates. Training and evaluation were performed in a square grid of 5 by 5 with 5 hunters and 5 rabbits. Results are averaged over 5 seeds. For each seed, training and evaluation were run for 1 million and 1 thousand episodes respectively.

Compared with no entropy, crude entropy can actually reduce performance. However, smoothed entropy and smoothed mode entropy always increase performance, often significantly. For the LSTM

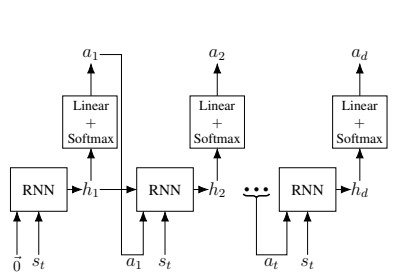

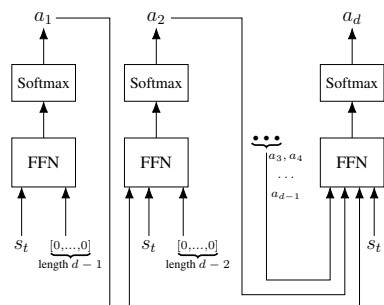

(a) The RNN architecture. To generate $a_i$, we input $s_t$ and $a_{i-1}$ into the RNN and then pass the resulting hidden state $h_i$ through a linear layer and a softmax to generate a distribution, from which we sample $a_i$.

(b) The MMDP architecture. To generate $a_i$, we input $s_t$ and $a_1, a_2, \ldots, a_{i-1}$ into a FFN. The output is passed through a softmax layer, providing a distribution from which we sample $a_i$. Since the input size of the FFN is fixed, when generating $a_i$, constants 0 serve as placeholders for $a_{i+1}, \ldots, a_{d-1}$ in the input to the FFN.

Figure 1: The RNN and MMDP architectures for generating parameterized policies.

model, the best performing approximation is smoothed entropy. We also note that there is not a significant difference in performance between the smoothed entropy, smoothed mode entropy, and the unbiased gradient approaches. When comparing the four models, we see that the LSTM model with smoothed entropy does significantly better the other three models. The CommNet model could potentially be improved by allowing the hunters to see more of the state; this could be investigated in future research.

Table 1: Performance of IS, LSTM, MMDP and CommNet across different entropy approximations.

|  | Without Entropy | Crude Entropy | Smoothed Entropy | Smoothed Mode Entropy | Unbiased Gradient Estimate |
|---|---|---|---|---|---|
| IS Mean Episode Length | $98.7 \pm 78.9$ | $32 \pm 12.3$ | $11.8 \pm 1.9$ | $11.8 \pm 1.9$ | $11.8 \pm 1.9$ |
| LSTM Mean Episode Length | $10.1 \pm 1.9$ | $19 \pm 8.7$ | $\mathbf{5.6 \pm 0.2}$ | $6.0 \pm 0.2$ | $6.0 \pm 0.1$ |
| MMDP Mean Episode Length | $21.5 \pm 3.7$ | $37.3 \pm 29.6$ | $10.6 \pm 0.7$ | $10.6 \pm 0.7$ | $9.8 \pm 0.6$ |
| CommNet Mean Episode Length | $22.7 \pm 0.6$ | $22.3 \pm 0.4$ | $21.9 \pm 0.4$ | $21.9 \pm 0.4$ | $21.9 \pm 0.4$ |
| IS Mean Episode Reward | $2.2 \pm 0.03$ | $2.4 \pm 0.05$ | $2.7 \pm 0.01$ | $2.7 \pm 0.01$ | $2.7 \pm 0.01$ |
| LSTM Mean Episode Reward | $3.0 \pm 0.06$ | $3.0 \pm 0.03$ | $\mathbf{3.3 \pm 0.04}$ | $3.2 \pm 0.04$ | $3.2 \pm 0.02$ |
| MMDP Mean Episode Reward | $2.8 \pm 0.03$ | $2.7 \pm 0.03$ | $2.9 \pm 0.03$ | $2.8 \pm 0.04$ | $2.9 \pm 0.02$ |
| CommNet Mean Episode Reward | $2.5 \pm 0.01$ | $2.6 \pm 0.01$ | $2.6 \pm 0.01$ | $2.6 \pm 0.01$ | $2.6 \pm 0.01$ |

In conclusion, we found that the smoothed estimate of the entropy and the unbiased estimate of the entropy gradient can significantly increase performance with marginal additional computational cost.

ACKNOWLEDGMENTS

We would like to thank Martin Arjovsky for his insightful comments. Our gratitude also goes to the HPC team at NYU, NYU Shanghai, and NYU Abu Dhabi.

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
