# OpenReview forum: "Efficient Entropy For Policy Gradient with Multi-Dimensional Action Space"
_ICLR.cc/2018/Workshop — Accept_

### Official Review · AnonReviewer1 · 2018-03-06
**Summary of paper that was on the edge for main conference**

**Rating:** 6
**Confidence:** 5

**Review:**

This paper is a short version of the paper "Policy Gradient For Multidimensional Action Spaces: Action Sampling and Entropy Bonus", which I was also a reviewer on. I gave the original paper a borderline accept rating and I'd be willing to give this paper the same rating.

---

### Official Review · AnonReviewer3 · 2018-03-11

**Rating:** 6
**Confidence:** 4

**Review:**

The paper proposes entropy estimates for autoregressive models for policy gradient.
I think, the proposed estimates are novel and interesting. The paper is well written and motivated.

The experiments are done on toy task, though I appreciate that the authors validated the proposed method as compared to various other baselines.

Also, as a side point,  the objective function can be formulated in terms of score function estimator, that might be worth pointing out in the paper.

---

### Decision · Program_Chairs · 2018-03-20
**ICLR 2018 Workshop Acceptance Decision**

**Decision:**

Accept

**Comment:**

Congratulations, your paper was accepted to the ICLR workshop.